# HALLUCINATIVE TOPOLOGICAL MEMORY FOR ZERO-SHOT VISUAL PLANNING

## ABSTRACT

In visual planning (VP), an agent learns to plan goal-directed behavior from observations of a dynamical system obtained offline, e.g., images obtained from self-supervised robot interaction. VP algorithms essentially combine data-driven perception and planning, and are important for robotic manipulation and navigation domains, among others. A recent and promising approach to VP is the semi-parametric topological memory (SPTM) method, where image samples are treated as nodes in a graph, and the connectivity in the graph is learned using deep image classification. Thus, the learned graph represents the topological connectivity of the data, and planning can be performed using conventional graph search methods. However, training SPTM necessitates a suitable loss function for the connectivity classifier, which requires non-trivial manual tuning. More importantly, SPTM is constricted in its ability to generalize to changes in the domain, as its graph is constructed from direct observations and thus requires collecting new samples for planning. In this paper, we propose Hallucinative Topological Memory (HTM), which overcomes these shortcomings. In HTM, instead of training a discriminative classifier we train an energy function using contrastive predictive coding. In addition, we learn a conditional VAE model that generates samples given a context image of the domain, and use these *hallucinated* samples for building the connectivity graph, allowing for zero-shot generalization to domain changes. In simulated domains, HTM outperforms conventional SPTM and visual foresight methods in terms of both plan quality and success in long-horizon planning.

## 1 INTRODUCTION

For robots to operate in unstructured environments such as homes and hospitals, they need to manipulate objects and solve complex tasks as they perceive the physical world. While task planning and object manipulation have been studied in the classical AI paradigm [20, 9, 30, 10], most successes have relied on a human-designed state representation and perception, which can be challenging to obtain in unstructured domains. While high-dimensional sensory input such as images can be easy to acquire, planning using raw percepts is challenging. This has motivated the investigation of data-driven approaches for robotic manipulation. For example, deep reinforcement learning (RL) has made impressive progress in handling high-dimensional sensory inputs and solving complex tasks in recent years [7, 4, 15, 23].

One of the main challenges in deploying deep RL methods in human-centric environment is interpretability. For example, before executing a potentially dangerous task, it would be desirable to visualize what the robot is planning to do step by step, and intervene if necessary. Addressing both data-driven modeling and interpretability, the *visual planning* (VP) paradigm seeks to learn a model of the environment from raw perception and then produce a visual plan of solving a task before actually executing a robot action. Recently, several studies in manipulation and navigation [13, 29, 5, 22] have investigated VP approaches that first learn what is possible to do in a particular environment by self-supervised interaction, and then use the learned model to generate a visual plan from the current state to the goal, and finally apply visual servoing to follow the plan.

One particularly promising approach to VP is the semi-parametric topological memory (SPTM) method proposed by Savinov et al. [22]. In SPTM, images collected offline are treated as nodes in a graph and represent the possible states of the system. To connect nodes in this graph, an image classifier is trained to predict whether pairs of images were 'close' in the data or not, effectively

learning which image transitions are feasible in a small number of steps. The SPTM graph can then be used to generate a visual plan – a sequence of images between a pair of start and goal images – by directly searching the graph. SPTM has several advantages, such as producing highly interpretable visual plans and the ability to plan long-horizon behavior.

However, since SPTM builds the visual plan directly from images in the data, when the environment changes – for example, the lighting varies, the camera is slightly moved, or other objects are displaced – SPTM requires *recollecting* images in the new environment; in this sense, SPTM *does not generalize in a zero-shot sense*. Additionally, similar to [5], we find that training the graph connectivity classifier as originally proposed by [22] requires extensive manual tuning.

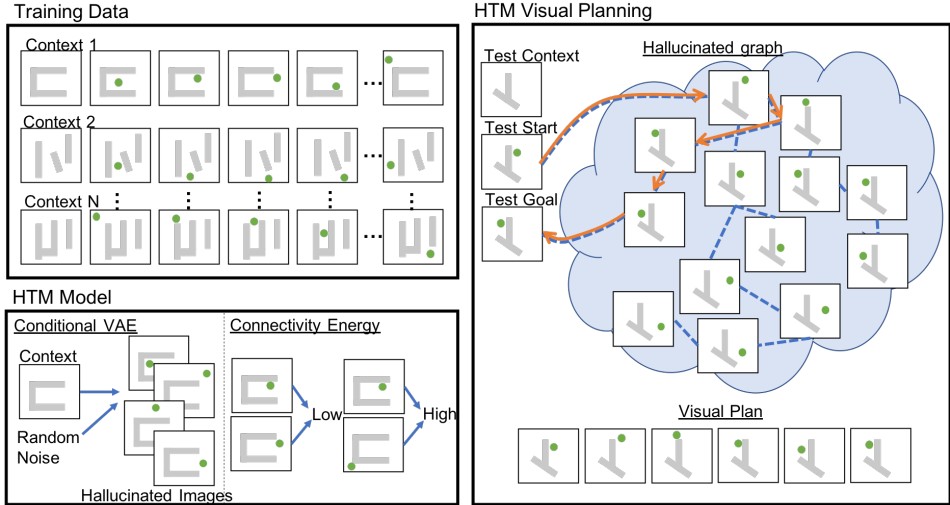

Figure 1: HTM illustration. **Top left:** data collection. In this illustration, the task is to move a green object between gray obstacles. Data consists of multiple obstacle configurations (contexts), and images of random movement of the object in each configuration. **Bottom left:** the elements of HTM. A CVAE is trained to hallucinate images of the object and obstacles conditioned on the obstacle image context. A connectivity energy model is trained to score pairs of images based on the feasibility of their transition. **Right:** HTM visual planning. Given a new context image and a pair of start and goal images, we first use the CVAE to hallucinate possible images of the object and obstacles. Then, a connectivity graph (blue dotted lines) is computed based on the connectivity energy, and we plan for the shortest path from start to goal on this graph (orange solid line). For executing the plan, a visual servoing controller is later used to track the image sequence.

In this work, we propose to improve both the robustness and zero-shot generalization of SPTM. To tackle the issue of generalization, we assume that the environment is described using some context vector, which can be an image of the domain or any other observation data that contains enough information to extract a plan (see Figure 1 top left). We then train a conditional generative model that hallucinates possible states of the domain conditioned on the context vector. Thus, given an unseen context, the generative model hallucinates exploration data without requiring actual exploration. When building the connectivity graph with these hallucinated images, we replace the vanilla classifier used in SPTM with an energy-based model that employs a contrastive loss. We show that this alteration drastically improves planning robustness and quality. Finally, for planning, instead of connecting nodes in the graph according to an arbitrary threshold of the connectivity classifier, as in SPTM, we cast the planning as an inference problem, and efficiently search for the shortest path in a graph with weights proportional to the inverse of a proximity score from our energy model. Empirically, we demonstrate that this provides much smoother plans and barely requires any hyperparameter tuning. We term our approach Hallucinative Topological Memory (HTM). A visual overview of our algorithm is presented in Figure 1.

We evaluate our method on a set of simulated VP problems of moving an object between obstacles, which require long-horizon planning. In contrast with prior work, which only focused on the success of the method in executing a task, here we also measure the *interpretability* of visual planning, through mean opinion scores of features such as image fidelity and feasibility of the image sequence.

In both measures, HTM outperforms state-of-the-art data-driven approaches such as visual foresight [4] and the original SPTM.

## 2 BACKGROUND

**Context-Conditional Visual Planning and Acting (VPA) Problem.** We consider the context-conditional visual planning problem from [13, 29]. Consider deterministic and fully-observable environments $\mathcal{E}_1, ..., \mathcal{E}_N$ that are sampled from an environment distribution $P_{\mathcal{E}}$. Each environment $\mathcal{E}_i$ can be described by a context vector $c_i$ that entirely defines the dynamics $o_{t+1}^i = m(o_t^i, a_t^i | c_i)$, where $o_t^i, a_t^i$ are the observations and actions, respectively, at timestep $t$ from context $c_i$. For example, in the illustration in Figure 1, the context could represent an image of the obstacle positions, which is enough to predict the possible movement of objects in the domain.[1] As is typical in VP problems, we assume our data $\mathcal{D} = \{o_1^i, a_1^i, ..., o_{T_i}^i, c_i\}_{i \in \{1, ..., N\}}$ is collected in a self-supervised manner, and that in each environment $\mathcal{E}_i$, the observation distribution is defined as $P_o(\cdot | c_i)$. At test time, we are presented with a new environment, its corresponding context vector $c$, and a pair of start and goal observations $o_{start}, o_{goal}$. Our goal is to use the training data to build a planner $Q_h(o_{start}, o_{goal}, c)$ and an h-horizon policy $\pi_h$. The planner's task is to generate a sequence of observations between $o_{start}$ and $o_{goal}$, in which any two consecutive observations are reachable within $h$ time steps. The policy takes as input the image sequence and outputs a control policy that transitions the system from $o_{start}$ to $o_{goal}$. As the problem requires a full plan given only a context image in the new environment, the planner must be capable of zero-shot generalization. Note that the planner and policy form an interpretable planning method that allows us to evaluate their performance separately. For simplicity we will omit the subscript $h$ for the planner and the policy.

**Semi-Parametric Topological Memory (SPTM)** [22] is a visual planning method that can be used to solve a special case of VPA. where there is only a single training environment, $\mathcal{E}$ and no context image. SPTM builds a memory-based planner and an inverse-model controller. At training, a classifier $R$ is trained to map two observation images $o_i, o_j$ to a score $\in [0, 1]$ representing the feasibility of the transition, where images that are $\leq h$ steps apart are labeled positive and images that are $\geq l$ are negative. The policy is trained as an inverse model $L$, mapping a pair of observation images $o_i, o_j$ to an appropriate action $a$ that transitions the system from $o_i$ to $o_j$.

Given an unseen environment $\mathcal{E}^*$, new observations are *manually* collected and organized as nodes in a graph $G$. Edges in the graph connect observations $o_i, o_j$ if $R(o_i, o_j) \geq s_{shortcut}$, where $s_{shortcut}$ is a manually defined threshold. To plan, given start and goal observations $o_{start}$ and $o_{goal}$, SPTM first uses $R$ to localize, i.e., find the closest nodes in $G$ to $o_{start}$ and $o_{goal}$. A path is found by running Dijkstra's algorithm, and the method then selects a waypoint $o_{w_i}$ on the path which represents the farthest observation that is still feasible under $R$. Since both the current localized state $o_i$ and its waypoint $o_{w_i}$ are in the observation space, we can directly apply the inverse model and take the action $a_i$ where $a_i = L(o_i, o_{w_i})$. After localizing to the new observation state reached by $a_i$, SPTM repeats the process until the node closest to $o_{goal}$ is reached.

**Conditional Variational Auto-Encoder (CVAE)** [25] is a deep generative model that can be used for learning a high-dimensional conditional distribution $P_o(\cdot | c)$. The CVAE is trained by maximizing the evidence lower bound (ELBO): $\mathcal{L}_{CVAE} = -D_{KL}(q_\theta(z|o, c) | r_\psi(z|c)) + \mathbb{E}_{q_\phi(z|o,c)}[\log p_\theta(o|z, c)]$, where $q_\theta(z|o, c)$ is the encoder that maps observations and contexts to the latent distribution, $p_\theta(o|z, c)$ is the decoder that maps latents and contexts to the observation distribution, and $r_\psi(z|c)$ is the prior that maps contexts to latent prior distributions. Together $p_\theta, q_\phi, r_\psi$ are trained to maximize the variational lower bound above. We assume that the prior and the encoder are Gaussian, which allows the $D_{KL}$ term to be computed in closed-form. Monte-Carlo sampling and the reparametrization trick [12] are used to approximate the gradient of the loss.

**Contrastive Predictive Coding (CPC)** [17] extracts compact representations that maximize the causal and predictive aspects of high-dimensional sequential data. A non-linear encoder $g_{enc}$ encodes the observation $o_t$ to a latent representation $z_t = g_{enc}(o_t)$. We maximize the mutual information between the latent representation $z_t$ and future observation $o_{t+k}$ with a log-

---

[1]We used such a context image in our experiments. We assume that in a practical application, observing the domain without the robot would be feasible, making this setting relevant to applications.

bilinear model[2] $f_k(o_{t+k}, o_t) = \exp(z_{t+k}^T W_k z_t)$. This model is trained to be proportional to the density ratio $p(o_{t+k}|z_t)/p(o_{t+k})$ by the CPC loss function: the cross entropy loss of correctly classifying a positive sample from a set $X = \{o_1, ..., o_N\}$ of $N$ random samples with 1 positive sample from $p(o_{t+k}|z_t)$ and $N - 1$ negative samples sampled from $p(o_{t+k})$: $\mathcal{L}_{CPC} =$

$$-\mathbb{E}_{o_t, o_{t+k}} \left[ \log \frac{f_k(o_{t+k}, o_t)}{\sum_{o_j \in X} f_k(o_j, o_t)} \right].$$

## 3 HALLUCINATIVE TOPOLOGICAL MEMORY

SPTM has been shown to solve long-horizon planning problems such as navigation from first-person view [22]. However, *SPTM is not zero-shot*: even a small change to the training environment requires collecting substantial exploration data for building the planning graph. This can be a limitation in practice, especially in robotic domains, as any interaction with the environment requires robot time, and exploring a new environment can be challenging (indeed, [22] applied manual exploration). In addition, similarly to [5], we found that training the connectivity classifier as proposed in [22] requires extensive hyperparameter tuning.

In this section, we propose an extension of SPTM to overcome these two challenges by employing three ideas – (1) using a CVAE [25] to hallucinate samples in a zero-shot setting, (2) using contrastive loss for a more robust score function and planner, and (3) planning based on an approximate maximum likelihood formulation of the shortest path under uniform state distribution. We call this approach Hallucinative Topological Memory (HTM), and next detail each component in our method.

### 3.1 HALLUCINATING SAMPLES WITH CVAE

We propose a zero-shot learning solution for automatically building the planning graph using only a context vector of the new environment. Our idea is that, after seeing many different environments and corresponding states of the system during training, given a new environment we should be able to effectively *hallucinate* possible system states. We can then use these hallucinations in lieu of real samples from the system in order to build the planning graph. To generate images conditioned on a context, we implement a CVAE as depicted in Figure 1. During training, we learn the prior latent distribution $r_\psi(z|c)$, modeled as a Gaussian with mean $\mu(c)$ and covariance matrix $\Sigma(c)$, where $\mu(\cdot)$ and $\Sigma(\cdot)$ are learned non-linear neural network transformations. During testing, when prompted with a new context vector $c$, we can sample latent vectors $z_1, ..., z_N \mid c \sim \mathcal{N}(\mu(c), \Sigma(c))$, and pass them through the decoder $p_\theta(x|z, c)$ for hallucinating samples in replacement of exploration data.

### 3.2 LEARNING THE CONNECTIVITY VIA CONTRASTIVE LOSS

A critical component in the SPTM method is the connectivity classifier that decides which image transitions are feasible. False positives may result in impossible short-cuts in the graph, while false negatives can make the plan unnecessarily long. In [22], the classifier was trained discriminatively, using observations in the data that were reached within $h$ steps as positive examples, and more than $l$ steps as negative examples, where $h$ and $l$ are chosen arbitrarily. In practice, this leads to three important problems. First, this method is known to be sensitive to the choice of positive and negative labeling [5]. Second, training data are required to be long, non-cyclic trajectories for a high likelihood of sampling 'true' negative samples. However, self-supervised interaction data often resembles random walks that repeatedly visit a similar state, leading to inconsistent estimates on what constitutes negative data. Third, since the classifier is only trained to predict positively for temporally nearby images and negatively for temporally far away images, its predictions of *medium-distance* images can be arbitrary. This creates both false positives and false negatives, thereby increasing shortcuts and missing edges in the graph.

To solve these problems, we propose to learn a connectivity score using contrastive predictive loss [17]. Similar to CVAE, we initialize a CPC encoder $g_{enc}$ that takes in both observation and context, and a density-ratio model $f_k$ that does not depend on the context. Through optimizing the CPC objective, $f_k$ of positive pairs are encouraged to be distinguishable from that of negative pairs. Thus, it serves as a proxy for the temporal distance between two observations, leading to a connectivity score for planning. Theoretically, CPC loss is better motivated than the classification loss in SPTM

---

[2]The original CPC model has an additional autoregressive memory variable [17]. We drop it in our formulation as our domains are fully observable and do not require memory.

as it structures the latent space on a clear objective: maximize the mutual information between current and future observations. In practice, this results in less hyperparameter tuning and a smoother distance manifold in the representation space. Finally, instead of only sampling from the same trajectory as done in SPTM, our negative data are collected by sampling from the latent space of a trained CVAE or the replay buffer. Without this trick, we found that the SPTM classifier fails to handle self-supervised data.

### 3.3 PLANNING AS INFERENCE

**Planning Algorithm.** Given a start observation $o_{start}$, a goal observation $o_{goal}$ sampled from a potentially new environment $\mathcal{E}^*$, and the context vector $c$, we propose a 4-step planning algorithm. First, we hallucinate exploration data by sampling from the latent space $P_o(\cdot|c)$ of the CVAE. Second, we build a fully-connected weighted graph $G(V, E)$ by forming connections between all $i$ generated image samples $\hat{o}_i = p_\theta(\cdot|z_i, c)$, where $p_\theta$ is the trained CVAE decoder and $z_i$ is the vector sampled from the CVAE prior. We choose our edge weight between nodes $i$ and $j$ as one of two choices: (1) an energy model, i.e., the inverse of $f_k(i, j)$, or (2) a density ratio, i.e., the inverse of normalized $f_k$ between two nodes, ie. $f_k(i,j)/\sum_l(f_k(i,l))$. This score reflects the difficulty in transitioning to the next state from the current state by self-supervised exploration. The learned connectivity graph $G$ can be viewed as a topological memory upon which we can use conventional graph planning methods to efficiently perform visual planning. In the third step, we find the shortest path using Dijkstra's algorithm on the learned connectivity graph $G$ between the start and end node. In the fourth step, we apply our policy to follow the visual plan, reaching the next node in our shortest path and replan every fixed number of steps until we reach $\hat{o}_{goal}$. For the policy, we train an inverse model which predicts actions given two observations that are within $h$ steps apart.

**Maximum likelihood trajectory with Dijkstra's.** We show that the CPC loss can be utilized to cast the planning problem as an inference problem, and results in an effective planning algorithm. After training the CPC objective to convergence, we have $f_k(o_{t+k}, o_t) \propto p(o_{t+k}|o_t)/p(o_{t+k})$ [17]. To estimate $p(o_{t+k}|o_t)/p(o_{t+k})$, we compute the normalizing factor $\sum_{o' \in V}[f_k(o', o_t)]$ for each $o_t$ by averaging over all nodes in the graph. Let's define our non-negative weight from $o_t$ to $o_{t+k}$ as $\omega(o_{t+k}, o_t) = \sum_{o' \in V}[f_k(o', o_t)]/f_k(o_{t+k}, o_t) \approx p(o_{t+k})/p(o_{t+k}|o_t)$.

A shortest-path planning algorithm finds $T, o_0, ..., o_T$ that minimizes $\sum_{t=0}^{T-1} \omega(o_t, o_{t+1})$ such that $o_0 = o_{start}, o_T = o_{goal}$. By Jensen's inequality and the Markovian property of $o_0, ..., o_T$ we have that, $\log \frac{1}{T}\sum_{t=0}^{T-1} \omega(o_t, o_{t+1}) \geq \frac{1}{T}\sum_{t=0}^{T-1} \log \omega(o_t, o_{t+1}) = \frac{1}{T}\sum_{t=0}^{T-1}(\log p(o_{t+1}) - \log p(o_{t+1}|o_t)) = \frac{1}{T}\sum_{t=1}^{T-1} p(o_t) - \log p(o_1, ..., o_{T-1}|o_0 = o_{start}, o_T = o_{goal})$, Thus, assuming that the self-supervised data distribution is approximately uniform, the shortest path algorithm with proposed weight $\omega$ maximizes a lower bound on the trajectory likelihood given the start and goal states. In practice, this leads to a more stable planning approach and yields more feasible plans.

## 4 RELATED WORK

**Reinforcement Learning.** Most of the study of data-driven planning has been under the model-free RL framework [23, 15, 24]. However, the need to design a reward function, and the fact that the learned policy does not generalize to tasks that are not defined by the specific reward, has motivated the study of model-based approaches. Recently, [11, 8] investigated model-based RL from pixels on Mujoco and Atari domains, but did not study generalization to a new environment. [6, 4] explored model-based RL with image-based goals using visual model predictive control (visual MPC). These methods rely on video prediction, and are limited in the planning horizon due to accumulating errors. In comparison, our method does not predict full trajectories but only individual images, mitigating this problem. Our method can also use visual MPC as a replacement for the visual servoing policy.

**Self-supervised learning.** Several studies investigated planning goal directed behavior from data obtained offline, e.g., by self-supervised robot interaction [1, 18]. Nair et al. [16] used an inverse model to reach local sub-goals, but require human demonstrations of long-horizon plans. Wang et al. [29] solve the visual planning problem using a conditional version of Causal InfoGAN [13]. However, as training GAN is unstable and requires tedious model selection [21], we opted for the CVAE-based approach, which is much more robust.

**Classical planning and representation learning.** In classical planning literature, task and motion planning also separates the high-level planning and the low-level controller [30, 27, 10]. In these works, domain knowledge is required to specify preconditions and effects at the task level. Our approach only requires data collected through self-supervised interaction.

Other studies that bridge between classical planning and representation learning include [13, 3, 2, 5]. These works, however, do not consider zero-shot generalization. While Srinivas et al. [26] and Qureshi et al. [19] learn representations that allow goal-directed planning to unseen environments, they require expert training trajectories. Ichter and Pavone [8] also generalizes motion planning to new environments, but require a collision checker and valid samples from test environments.

## 5 EXPERIMENTS

Recent work in visual planning (e.g., [13, 29, 4]) focused on real robotic tasks with visual input. While impressive, such results can be difficult to reproduce or compare. For example, it is not clear whether manipulating a rope with the PR2 robot [29] is more or less difficult than manipulating a rigid object among many visual distractors [4]. In light of this difficulty, we propose a suite of simulated tasks with an explicit difficulty scale and clear evaluation metrics. Our domains consider moving a rigid object between obstacles using Mujoco [28], and by varying the obstacle positions, we can control the planning difficulty. For example, placing the object in a cul-de-sac would require non-trivial planning compared to simply moving around an obstacle along the way to the goal. We thus create two domains, as seen in Figure 2:

1. **Block wall:**: A green block navigates around a static red obstacle, which can vary in position.
2. **Block wall with complex obstacle**: Similar to the above, but here the wall is a 3-link object which can vary in position, joint angles, and length, making the task significantly harder.

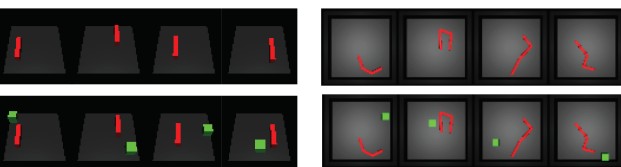

Figure 2: Block wall domain (left) and block wall with complex obstacle (right) domain. The top row shows some example contexts. The bottom rows show example observations.

With these domains, we aim to asses the following attributes:

- Does HTM improve **visual plan quality** over state-of-the-art VP methods [22, 4]?
- How does HTM **execution success rate** compare to state-of-the-art VP methods?
- How well does HTM **generalize** its planning to unseen contexts?

We discuss our evaluation metrics for these attributes in Section 5.1. To fully assess success of HTM relative to other state-of-the-art VP methods, we run these evaluation metrics on SPTM [22] and Visual Foresight [4]. In the first baseline, since vanilla SPTM cannot plan in a new environment, we use the same samples generated by the same CVAE as HTM, and then build the graph by assigning edge weights in the graph proportional to their exponentiated SPTM classifier score. [3] We also give it the same negative sampling proceedure as HTM. The same low-level controller is also used to follow the plans. In the second baseline, Visual Foresight trains a video prediction model, and then performs model predictive control (MPC) which finds the optimal action sequence through random shooting. For the random shooting, we used 3 iterations of the cross-entropy method with 200 sample sequences. The MPC acts for 10 steps and replans, where the planning horizon $T$ is 15. We use the state-of-the-art video predictor as proposed by Lee et al. [14] and the public code provided by the authors. For evaluating trajectories in random shooting, we studied two cost functions that are suitable for our domains: pixel MSE loss and green pixel distance. The pixel MSE loss computes the

---

[3]We found that negative exponential weighting, which requires no tuning, performed slightly better in results than our best tuned version of the original SPTM edge weighting through thresholding.

pixel distance between the predicted observations and the goal image. This provides a sparse signal when the object pixels in the plan can overlap with those of the goal. We also investigate a cost function that uses prior knowledge about the task – the position of the moving green block, which is approximated by calculating the center of mass of the green pixels. As opposed to pixel MSE, the green pixel distance provides a smooth cost function which estimates the normalized distance between the estimated block positions of the predicted observations and the goal image. Note that this assumes additional domain knowledge compared to HTM.

## 5.1 Evaluation Metrics

We design a set of tests that measure both qualitative and quantitative performance of an algorithm. To motivate the need for qualitative metrics, we reiterate the importance of planning interpretability; it is highly desirable that the generated plan visually make sense so as to allow a human to approve of the plan prior to execution.

**Qualitative** Visual plans have the essential property of being intuitive, in that the imagined trajectory is perceptually sensible. Since these qualities are highly subjective, we devised a set of tests to evaluate plans based on human visual perception. For each domain, we asked 5 participants to visually score 5 randomly generated plans from each model by answering the following questions: (1) *Fidelity*: Does the pixel quality of the images resemble the training data?; (2) *Feasibility*: Is each transition in the generated plan executable by a single action step?; and (3) *Completeness*: Is the goal reachable from the last image in the plan using a single action? Answers were in the range [0,1], where 0 denotes *No* to the proposed question and 1 means *Yes*. The mean opinion score were calculated for each model.

**Quantitative** In addition to generating visually sensible trajectories, a planning algorithm must also be able to successfully navigate towards a predefined goal. Thus, for each domain, we selected 20 start and goal images, each with an obstacle configuration unseen during training. Success was measured by the ability to get within some L2 distance to the goal in a $n$ steps or less, where the distance threshold and $n$ varied on the domain but was held constant across all models. A controller specified by the algorithm executed actions given an imagined trajectory, and replanning occurred every $r$ steps. Specific details can be found in the Appendix D.

## 5.2 Results

As shown in Table 5.2, HTM outperforms all baselines in both qualitative and quantitative measurements across all domains. In the simpler block wall domain, Visual Foresight with green pixel distance only succeeds under the assumption of additional state information of the object's location. the other algorithms do not have. However, in the complex obstacle domain, Visual Foresight fails to perform comparably to our algorithm, regardless of the additional assumption. We also compared our method with SPTM, using the same inverse model and CVAE to imagine testing samples. However, without a robust classification loss and improved method of weighting the graph's edges, SPTM often fails to find meaningful transitions.

In regards to perceptual evaluation, Visual Foresight generates realistic transitions, as seen by the high participant scores for feasibility. However, the algorithm is limited in creating a visual plan within the optimal $T = 15$ timesteps. [4] Thus, when confronted with a challenging task of navigating around a convex shape where the number of timesteps required exceeds $T$, Visual Foresight fails to construct a reliable plan (see Figure 3), and thus lacks plan completeness. Conversely, SPTM is able to imagine some trajectory that will reach the goal state. However, as mentioned above and was confirmed in the perceptual scores, SPTM fails to select feasible transitions, such as imagining a trajectory where the block will jump across the wall or split into two blocks. Our approach, on the other hand, received the highest scores of fidelity, feasibility, and completeness. Finally, we show in Figure 3 the results of our two proposed improvements to SPTM in isolation. The results clearly show that a classifier using contrastive loss outperforms that which uses Binary Cross Entropy (BCE) loss, and furthermore that the inverse of the score function for edge weighting is more successful than the best tuned version of binary edge weights.

---

[4]For plans require $> T$ steps, we found that error across the image translations accumulate and the predicted image drastically decreases in interpretability. This optimal value of $T$ is consistent with that of [4].

| Algorithms | Domain | Fidelity | Feasibility | Completeness | Execution Success |
|---|---|---|---|---|---|
| **HTM (1)** | 1 | **0.86 ± .05** | **0.84 ± .16** | **1.00 ± .00** | **100%** |
| | 2 | **0.95 ± .03** | **0.92 ± .11** | **1.00 ± .00** | **95%** |
| **HTM (2)** | 1 | **0.75 ± .09** | **0.88 ± .14** | **1.00 ± .00** | **95%** |
| | 2 | **0.96 ± .03** | **0.96 ± .08** | **0.96 ± .08** | **100%** |
| SPTM with CVAE | 1 | 0.40 ± .11 | 0.00 ± .00 | 1.00 ± .00 | 55% |
| | 2 | 0.92 ± .07 | 0.00 ± .00 | 1.00 ± .00 | 30% |
| Visual Foresight [4] (pixel MSE loss) | 1 | 0.74 ± .08 | 0.84 ± .16 | 0.04 ±. 08 | 25% |
| | 2 | 0.59 ± .16 | 0.64 ± .21 | 0.00 ± .00 | 0% |
| Visual Foresight [4] (green pixel distance) | 1 | 0.80 ± .07 | 0.84 ± .16 | 0.04 ± .08 | 90% |
| | 2 | 0.69 ± .14 | 0.56 ± .21 | 0.00 ± .00 | 35% |
| Inverse Model | 1 | - | - | - | 20% |
| | 2 | - | - | - | 25% |

Table 1: Qualitative and quantitative evaluation for the the block wall (1) and block wall with complex obstacle (2) domains. Qualitative data also displays the 95% confidence interval. Note HTM (1) refers to edge weighting using the energy model, and (2) is weighting using the density ratio, as described in 3.3.

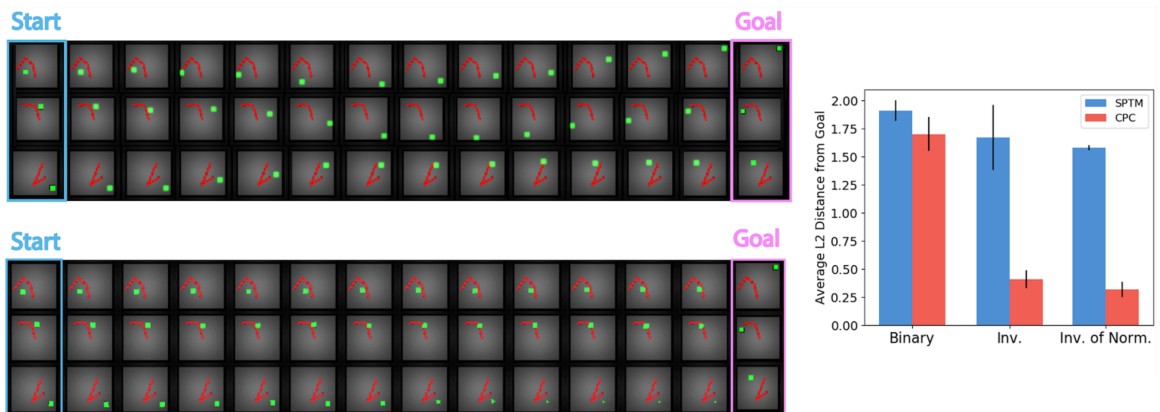

Figure 3: **Left:** HTM plan examples (top) and Visual Foresight (green pixel distance) plan examples (bottom). Note Visual Foresight is unable to conduct a long-horizon plan, and thus greedily moves in the direction of the goal state. **Right:** Comparison of score function and edge weighting function by examining final *average distance* to the goal state for 10 test start/goal pairs (the lower the better). For the score function, we denote the energy model structured with contrastive loss as *CPC* and the classifier as proposed in [22] with BCE loss as *SPTM*. For the edge weighting function, we test the binary edge weighting from the original SPTM paper, the inverse of the score function, and the inverse of the normalized score function.

## 6    DISCUSSION

We propose a method that is visually interpretable and modular – we first hallucinate possible configurations, then compute a connectivity between them, and then plan. Our HTM can generalize to unseen environments and improve visual plan quality and execution success rate over state-of-the-art VP methods. Our results suggest that combining classical planning methods with data-driven perception can be helpful for long-horizon visual planning problems, and takes another step in bridging the gap between learning and planning. In future work, we plan to combine HTM with Visual MPC for handling more complex objects, and use object-oriented planning for handling multiple objects. Another interesting aspect is to improve planning by hallucinating samples conditioned on the start and goal configurations, which can help reduce the search space during planning.

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

## A    DISCRIMINATIVE MODELS: CLASSIFIER VS. ENERGY MODEL

In this section, we assume the dataset as described in VPA, $\mathcal{D} = \{o_1^i, ..., o_{T_i}^i\}_{i=1}^n$. There are two ways of learning a model to distinguish the positive from the negative transitions.

**Classifier:** As noted above, SPTM first trains a classifier which distinguishes between an image pair that is within $h$ steps apart, and the images that are far apart using random sampling. The classifier is used to localize the current image and find possible next images for planning. In essence, the classifier contains the encoder $g_\theta$ that embeds the observation $x$ and the the score function $f$ that takes the embedding of each image and output the logit for a sigmoid function. The binary cross entropy loss of the classifier can be written as follows:

$$L_{SPTM}(\theta, \psi; \mathcal{D}) = - \sum_{(z_t, z_{t+k}) \sim \mathcal{D}} \left[ \log \frac{f(z, z_{t+k})}{1 + f(z_t, z_{t+k})} + \log \frac{1}{1 + f(z_t, z_t^-)} \right]$$

$$= - \sum_{(z_t, z_{t+1}) \sim \mathcal{D}} \log \left[ \frac{f_\psi(z_t, z_{t+k})}{1 + f_\psi(z_t, z_{t+k}) + f_\psi(z_t, z_t^-) + f_\psi(z_t, z_{t+k}) * f_\psi(z_t, z_t^-)} \right],$$

where $z_t^-$ is a random sample from $\mathcal{D}$.

**Energy model:** Another form of discriminating the the positive transition out of negative transitions is through an energy model. Oord et al. [17] learn the embeddings of the current states that are predictive of the future states. Let $g$ be an encoder of the input $x$ and $z = g_\theta(x)$ be the embedding. The loss function can be described as a cross entropy loss of predicting the correct sample from $N + 1$ samples which contain 1 positive sample and $N$ negative samples:

$$L_{CPC}(\theta, \psi; \mathcal{D}) = - \sum_{(z_t, z_{t+k}) \sim \mathcal{D}} \log \left[ \frac{f_\psi(z_t, z_{t+k})}{f_\psi(z_t, z_{t+k}) + \sum_{i=1}^N f_\psi(z_t, z_t^{i-})} \right],$$

where $f_\psi(u, v) = \exp(u^T \psi v)$ and $z_t^{1-}, ..., z_t^{N-}$ are the random samples from $\mathcal{D}$.

Note that when the number of negative samples is 1 the loss function resembles the SPTM.

## B    MUTUAL INFORMATION (MI)

This quantity measures how much knowing one variable reduces the uncertainty of the other variable. More precisely, the mutual information between two random variables $X$ and $Y$ can be described as $I(X, Y) = H(X) - H(X|Y) = H(Y) - H(Y|X) = \mathbb{E}_{X,Y} \left[ \frac{p_{X,Y}}{p_X p_Y} \right]$.

## C    ADDITIONAL PROOF

**Lemma C.1.** *For any random variable $X, Y$ and a deterministic function $g$, $I(X, Y) \geq I(X, g(Y))$.*[5]
*Proof.* $I(X, Y) = H(X) - H(X|Y) = H(X) - H(X|Y, g(Y)) \geq H(X) - H(X|g(Y)) = I(X, g(Y))$ □

## D    ADDITIONAL RESULTS AND HYPERPARAMETERS

---

[5]This is also known as the *data processing inequality*.

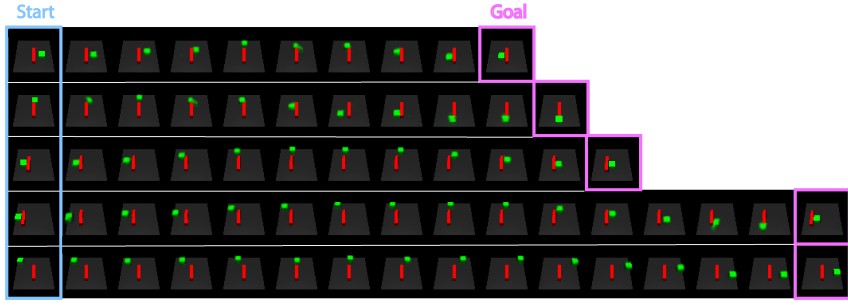

Figure 4: HTM plan examples. The hallucination allows the planner to imagine how to go around the wall even though it has not seen the context before.

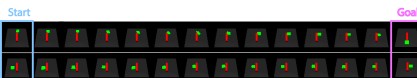

Figure 5: Visual Foresight plan examples. The plans do not completely show the trajectory to the goal.

Table 2: Data parameters.

|  | Domain 1 | Domain 2 |
|---|---|---|
| no. contexts | 150 | 400 |
| initializations per context | 50 | 30 |
| trajectory length | 20 | 100 |
| action space | $U[-.05, .05]$ | $U[-.1, .1]$ |

Table 3: Planning hyperparameters.

|  | Domain 1 | Domain 2 |
|---|---|---|
| no. of samples from CVAE | 300 | 500 |
| L2 threshold for success | .5 | .75 |
| $n$ (timesteps to get to goal) | 500 | 400 |
| $r$ (timesteps until replanning) | 200 | 80 |

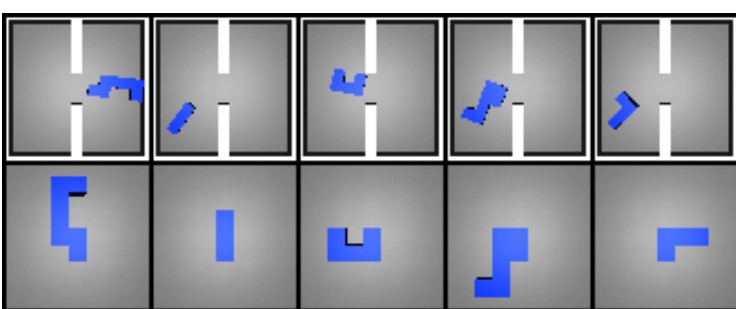

Figure 6: Sample observations (top) and contexts (bottom). In this domain, an object can be translated and rotated (SE(2)) slightly per timestep. The data are collected from 360 different object shapes with different number of building blocks between 3 to 7. Each object is randomly initialized 50 times and each episode has length 30. The goal is to plan a manipulation of an *unseen* object through the narrow gap between obstacles in zero-shot.

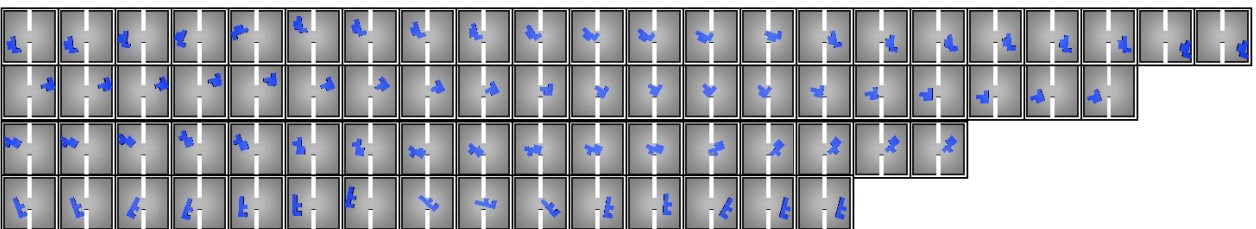

Figure 7: HTM evaluated on real data. Ground truth start and goal are the leftmost and rightmost images, respectively, in the row.

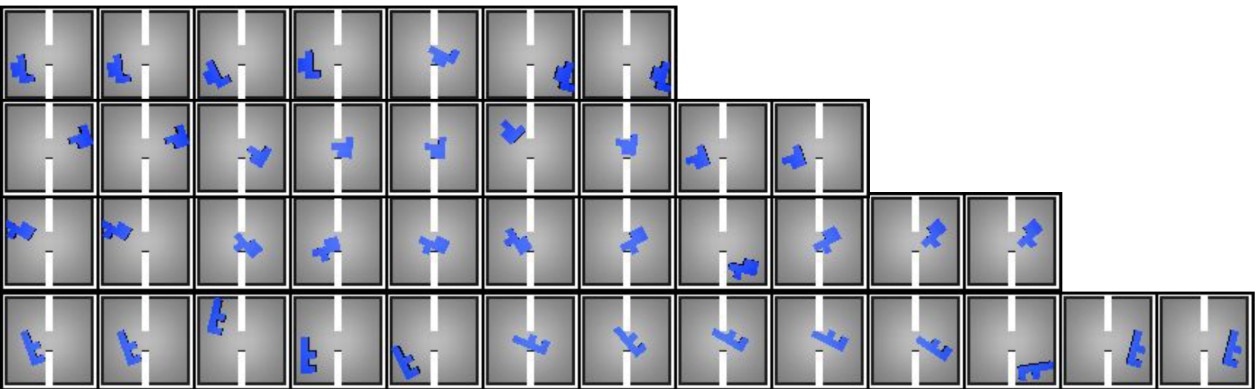

Figure 8: SPTM evaluated on real data. Ground truth start and goal are the leftmost and rightmost images, respectively, in the row.

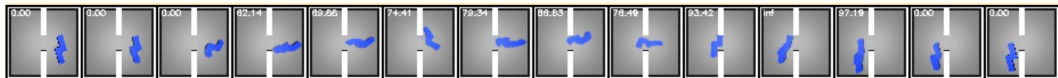

Figure 9: HTM evaluated on hallucinated data. Ground truth start and goal are the leftmost and rightmost images, respectively, in the row.

