# OpenReview forum: "Hallucinative Topological Memory for Zero-Shot Visual Planning"
_ICLR.cc/2020/Conference — Reject_

### Official Review · AnonReviewer1 · 2019-10-08
**Official Blind Review #1**

**Rating:** 1

**Review:**

The paper presents a method for learning agents to solve visual planning, in particular to navigate to a desired goal position in a maze, with a learned topological map, i.e. a graph, where nodes correspond to positions in the maze and edges correspond accessibility (reachability in a certain number of steps). The work extends previous work (semi parametric topological memory, ref. [22]) in several ways. It claims to address a shortcoming of [22], namely the fact that the graph is calculated offline from random rollouts, by using a conditional variational auto-encoder to predict a set of observed images which could lie between the current position and the goal position, and, most importantly from a context image which describes the layout of the environment. These predicted images are then arranged in a graph through a connectivity predictor, which is trained from rollouts through a contrastive loss. Training is performed on multiple environments, and the context vector provides enough information for this connectivity network to generalize to unseen environments.  At test time, the agent navigates using a planner and a policy. The planner calculates the shortest path on a graph where edges are connectivity probabilities, and the policy is an inverse model trained on the output of the planner.

While the idea of a topological memory with dynamic graph creation is certainly interesting, the work is unfortunately not well enough executed and the paper structured written in a way which makes it up to impossible to grasp what has been really done, as much information is missing which would be required for understanding.

As a first example, we are never really told what the observations are, which the agent sees. The different figures of the paper show very small images with a 2D maze from a bird’s eye view consisting of a walls arranged in a single connected component (mostly 1 to 3 strokes) in red color and an agent shown as a position indicated as a green dot. Are these the observed images? In absence of any other information, this is what we need to assume, and then this problem is fully observable and does not seem to be very challenging. Given the figures, even a handcrafted algorithm should be able to calculate the optimal solution with Dijkstra’s algorithm on a graph calculated from the pixel grid.

This important missing information alone makes it difficult to assess the paper, but the rest of the writing is similarly confusing. The authors focus on very short and dense descriptions of mathematical properties, but seem to have forgotten to ground the different symbols and to connect them to physical entities of the problem. The technical writing is in large parts disconnected from the problem which is addressed by it.

Further examples are:

-	“the data (…) is collected in a self-supervised manner”: what does this mean? Self-supervision is way of creating loss from data without labels, but I am not sure what is meant by collecting data this way.
-	The paragraph on CPC in section 2 can only be understood if the contrastive loss is known. To make the paper self-consistent, this should be properly explained, and tied to a training procedure which details how exactly the positive and negative samples are defined … and collected.
-	The CPC objective in section 2 is only loosely connected to its usage in section 3.2. Barely writing “we optimize a CPC” objective is not sufficient for understanding how this objective is really tied to the different entities of the problem. This paper contains maths (which is always a pleasure to read), but it is not a purely and abstract mathematical problem - a real task is addressed, so it needs to be connected to it. This connection has certainly been done by the authors while they were working on the problem, but they should also communicate it to the reader.
-	The section on ML trajectory is too dense and should be rewritten. I don’t understand what the authors want to tell us here. Basically, a (generalized) Dijkstra is run on a graph, where edge weights are the density or density ratios learned by the CPC objective, and if the edge weights are probabilities, that the shortest path corresponds to a trajectory likelihood. This is known, and this information is buried in a dense set of equations which are difficult to decipher and do not add any further value to the paper.
-	The connection between the planner (generalized Dijkstra) and the policy is never explained. We don’t know how the policy is trained and how it works.

One of the downsides of the method is that it requires a context image. This image is responsible for the generalization to unseen environments, but it is a major drawback, as the image must be created beforehand. The authors claim that the context image must only contain the layout in any format which makes it possible to extract information about navigational space from it, but in the experiments the context image corresponds to the full map – and it is probably equivalent to the observed images, but we can’t be sure as we haven’t been told. In any case, it is far from sure how this could generalize to more complex environments, let alone 3D navigation as is currently addressed in standard simulators like VizDoom, GIBSON, Matterport, Deepmind Lab, Habitat AI etc.

The authors’ claim that the proposed environment requires long-term planning, but looking at the images this does not seem to be the case.

The paper claims to perform zero-shot generalization and to adapt to changes in the environment, like the slight changes in camera motion, variations in lightning, but it unclear how the solution solves this claim.

How does the agent determine that a goal has been reached, without ground truth information?

What happens, if the hallucinated images are disconnected (form several connected components) or are disconnected from the current position and/or from the goal position?

As mentioned, the method is evaluated on an environment, which is too simple. The experiments are difficult to assess, as we don’t really know what the agent observes. An information asymmetry is mentioned (visual foresight having the object’s (=agent’s) position and the others not) … but if the proposed method observes the bird’s eye view, it can infer the agent’s position (as the position of the green dot).

Subjective evaluation by humans on this kind of simple data does not seem to be meaningful, in particular with a very low number of observers (5 people).


**Experience Assessment:**

I have published in this field for several years.

**Review Assessment: Checking Correctness Of Derivations And Theory:**

I carefully checked the derivations and theory.

**Review Assessment: Checking Correctness Of Experiments:**

I assessed the sensibility of the experiments.

**Review Assessment: Thoroughness In Paper Reading:**

I read the paper thoroughly.

---

> ### Author Response · Authors · 2019-11-15
> **Response to Reviewer #1 (Part 1/2)**
>
> Dear Reviewer #1,
>
> We appreciate your effort reading, reviewing, and giving us valuable feedback.
>
> Our paper tackles the visual planning problem: given images from a dynamical system, learn to predict goal-conditioned image trajectories. This problem has been studied recently in several settings, e.g., SPTM for 3D navigation [22], CIGAN and Visual-MPC for robotic manipulation [1,4,6,13,29].
>
> Here we extend SPTM, but *we do not consider 3D navigation*. Our observations are simply the overhead view images as plotted in the paper. These simple domains allow us to easily assess the capabilities of VP algorithms.
> While a navigation policy can indeed be manually coded using the pixel values, *this requires prior knowledge about the task!* - i.e., knowing that this is navigation between obstacles, knowing how obstacles look like, and knowing how to map the image to a relevant state space, action space, and configuration space. Our algorithm, and any other algorithm in the visual planning setting, *does not require any such knowledge in advance!*
>
> While these problems are indeed simple, and the planning horizons are relatively short, *state of the art VP methods cannot solve them*! This clearly demonstrates the (in-) capabilities of current VP methods, and demonstrates how difficult the visual planning problem really is.
>
> We kindly ask the reviewer to re-evaluate our paper *in the context of its contribution to the study of visual planning algorithms*, and not for general navigation problems.
>
> To further motivate our claims, we have added an additional experiment to show the generality of our approach. In this experiment, an object can be translated and rotated (in SE(2)) slightly per timestep. Different from the experiments in the paper, here we change the object shape (and give the object shape as a context image). The goal is to plan a manipulation of an unseen object through a narrow gap between obstacles in zero-shot. Thus, our algorithm has to learn how object geometry is related to movement between obstacles, and has to generalize this knowledge for planning. Of course, all this has to be learned only from raw images - we use exactly the same algorithm, regardless of the fact that now the state space, action space, and configuration space are different. Please see our general comment for more details and additional results.
>
> Paper writing: We acknowledge that our writing should be improved, and we thank the reviewer for pointing out many unclear points in the paper. We promise to write the paper in a much more accessible format, and clearly relate the math to the application. We emphasize that this can easily be done, and in the following we address the specific unclear points.
>
> Self-supervised data collection: Following work on self-supervised learning in robotics [1a, 2a, 3a], we mean that data is collected by letting the agent randomly explore the world without any specific supervision/guidance. Thus, we *do not* assume walkthroughs or demonstrations of any task, and the data only contains image sequences without any prior knowledge about the agent or the task. This is similar in spirit to the setting of off-policy batch RL, where the data collection policy is different from the learned behavior policy.
>
> Positive/negative data samples: Given the image sequence data, we use 1-step transitions from the data as positive samples, and random pairs of images from the data (collected under the same context image) as negative samples.
>
> CPC objective in sections 2, 3.2: We will clarify (including math) exactly how we use CPC on our data. As described above, by training on positive image pairs from the data and negative random pairs, the CPC loss learns to assign low energy to feasible transitions and high energy to infeasible transitions. This lets us use it as a proxy for connectivity in the graph.
>
> ML trajectory: The main point here is that the CPC output of transition probability (i.e., the energy of the transition) is not normalized. We propose a simple way to normalize it, and this makes interpreting the shortest path as an ML trajectory correct. Thus, the ML interpretation is not novel, but relating it to CPC is.
>
> Connection between planner and policy: The policy is a simple inverse model. It is trained by supervised learning to predict the action needed to bring the current state to the next state on the trajectory dataset. We use this inverse model to track the hallucinated sequence of images outputted by HTM. The same method was used in [29] using a different visual planning algorithm.

---

> > ### Author Response · Authors · 2019-11-15
> > **Response to Reviewer #1 (Part 2/2)**
> >
> > Context image: The context can in principle be a scene image, camera angles, lightning variables, or any other observation that contains information about the configuration space in the domain. While our experiment are very simple, we found that even this setting of giving the context as the full map is *very challenging* for visual planning methods, so in this work we did not experiment with more complex context variables.
> >
> > We point the reviewer to our new experiments, where the context there contains the agent shape and not the obstacle configuration, further demonstrating the generality of our approach.
> >
> > Long term planning: Moving around an obstacle definitely requires longer horizon planning than pushing an object without obstacles, as in [1,4,6]. That said, investigating visual planning for even longer horizon plans is an important future direction, which our work here gives even better motivation to study.
> >
> > Determining that goal has been reached: That’s a good point. In our simulated experiments, we know the groundtruth distance, and use that to stop the policy. In a real-world application, we would use other measures, such as pixel-distance or an image classifier trained to predict task success.
> >
> > Disconnected hallucinated images: Our method indeed builds a *fully connected* graph whose weights are the inverse of the normalized score function. High weights in the graph can effectively act as disconnection between nodes. Our experiments show that, at least in the domains investigated, our hallucination method is expressive enough to imagine enough diverse images to always find a smooth connected path.
> >
> > Our method explicitly prevents disconnected hallucinated images. Instead of computing edge-weights as binary value if the classifier score is above a certain threshold (as in SPTM), our method creates a *fully-connected graph*, so this is never an issue.
> >
> > Human evaluation: We found that the variance is quite small among the 5 testers. However, we are happy to add more human subjects to the evaluation if the reviewer finds it important. We also attached a link to the planning comparison examples sent to all participants for evaluation, which shows significant distinction in HTM planning results: https://tinyurl.com/htm-visualplan.
> >
> > As a final note, please observe that Figure 3 in our submission, which displays the visual plans of our algorithm against a baseline, was incorrect, and we have since posted the correct figure as a response several weeks ago (10/18). We also updated the PDF submission to reflect this.
> >
> > We thank you again for the questions to improve our paper. We take each of them seriously and will update the paper accordingly. We hope that our response also help clarifying the paper contributions.
> > In the context of visual planning algorithms, our paper (1) elucidates the difficulty of the problem in simple and easy to reproduce domains, (2) proposes a novel visual planning method, and (3) clearly demonstrates the benefits of the new method compared to previous state of the art. We kindly ask to re-evaluate our work in this context.
> >
> > Additional References:
> > [1a] Pinto, L. and Gupta, A., Supersizing self-supervision: Learning to grasp from 50k tries and 700 robot hours. In ICRA 2016.
> > [2a] Nair, A., Chen, D., Agrawal, P., Isola, P., Abbeel, P., Malik, J. and Levine, S., Combining self-supervised learning and imitation for vision-based rope manipulation. In ICRA 2017.
> > [3a] Ebert, F., Finn, C., Lee, A.X. and Levine, S., Self-supervised visual planning with temporal skip connections. arXiv preprint arXiv:1710.05268, 2017.

---

### Official Review · AnonReviewer2 · 2019-10-21
**Official Blind Review #2**

**Rating:** 8

**Review:**

The paper propose a novel visual planning approach which constructs explicit plans from "hallucinated" states of the environment. To hallucinate states, it uses a Conditional Variational Autoencoder (which is conditioned on a context image of the domain). To plan, it trains a Contrastive Predictive Coding (CPC) model for judging similarities between states, then applies this model to hallucinated states + start/end states, then runs Dijkstra on the edges weighted by similarities.

I vote for accepting this paper as it tackles two important problems: where to get subgoals for visual planning and what similarity function to use for zero-shot planning. Furthermore, the paper is clearly written, the experiments are well-conducted and analyzed.

Detailed arguments:
1. Where to get subgoals for visual planning is an important question persistently arising in control tasks. SPTM-style solution is indeed limited because it relies on an exploration sequence as a source of subgoals. Every time the environment changes, data would need to be re-collected. Getting subgoals from a conditional generative model is a neat solution.
2. Benchmarking similarity functions is crucial. One productive way to approach zero-shot problems is to employ similarity functions, but the question arises: what algorithm to use for training them? The paper compares two popular choices: CPC and Temporal Distance Classification (in particular, R-network). It thus provides guidance that CPC might be a better algorithm for training similarity functions.
3. The paper is well-positioned in the related work and points to the correct deficiencies of the existing methods. It also features nice experimental design with controlled complexity of the tasks, ablation studies and two relevant baselines.

I would encourage the authors to discuss the following questions:
1) Fidelity in Table 3 - why is it lower for SPTM compared to HTM if both methods rely on the same generated samples? Is it because HTM selects betters samples than SPTM for its plans?
2) Why is fidelity larger for SPTM in a more complex task 2?
3) Same question about fidelity/feasibility for HTM1/2?
4) Are there any plans to open-source the code?

**Experience Assessment:**

I have published one or two papers in this area.

**Review Assessment: Checking Correctness Of Derivations And Theory:**

I assessed the sensibility of the derivations and theory.

**Review Assessment: Checking Correctness Of Experiments:**

I carefully checked the experiments.

**Review Assessment: Thoroughness In Paper Reading:**

I read the paper thoroughly.

---

> ### Author Response · Authors · 2019-11-15
> **Response to Reviewer #2**
>
> Dear Reviewer #2,
>
> We are happy to hear that you enjoy our paper and appreciate our contributions. Thank you for your effort in reviewing the paper.
>
> (1) This is a great question, and you are exactly correct. Using the same pool of generated samples, we have found CPC to be more robust against poor generated images, e.g., irregular-shaped block, or double blocks, which give poor scores. On the other hand, a regular SPTM classifier tends to exploit these. We believe that this is due to the fact that the CPC estimates the mutual information which should be lower when the data are poorer. The CPC loss contrasts each positive pair against many negative pairs which make the score function more robust.
>
> (2)-(3) To simplify the learning problem, we modified the decoder architecture for domain 2 such that it learns a mask for combining the context and the CVAE output at the last layer. This helps reduce the number of parameters, speed up the training, and improve our sample quality. With better quality of image samples in the second domain, our algorithm was able to select from a larger pool of more realistic transitions, which elucidates the improvement in feasibility and fidelity scores. In conclusion, due to architecture differences, comparison between domains 1 and 2 is not fair, but comparison between different methods on the same domain is fair.
>
> (4) Yes. We plan to open-source the code for our algorithm, visual planning baselines, and the domains.

---

### Official Review · AnonReviewer3 · 2019-10-23
**Official Blind Review #3**

**Rating:** 6

**Review:**

The paper presents HTM, an extension of the semiparametric topological memory method that augments the approach with hallucinated nodes and an energy cost function. The hallucination is enabled by a CVAE, conditioned on an image of the environment, and allows the method to generalize to unseen environments. The energy cost function is trained as a contrastive loss and acts as a robustness score for connecting the two samples. The underlying graph is then used to plan for several top view planning problems.

The paper is well written and clear. I believe such latent representations are an interesting approach to solving visual navigation and general planning. HTM provides an interesting and useful extension to SPTM, allowing both generalization to unseen environments and a more robust loss function. The

My primary concern is the lack of rigorous experimentation to validate the concept and push it’s limits. The results in Table 1 show HTM outperforms baselines clearly on the given problems, but how it performs on more complex problems is unclear. These problems are dynamically simple and the obstacles are easily identified. Some more difficult problems may be:
- The mazes in SPTM or environments from https://arxiv.org/pdf/1612.03801.pdf.
- The original SPTM paper focuses on visual navigation from first person views. How does this method apply to such situations? How does the context translate to this scenario?
- Planning in real environments with real images, as done in [6].

Other comparisons and notes:
- Can the method be applied to higher dimensional problems (dimensionality of the underlying space) where planning may be more difficult? E.g. SE(2), robot arms or other agents from UPN [26]. Application with actual 3D workspace problems too would be interesting as the image context may underspecify the environment.
- The energy cost function acts as a proxy for connection probability when traversing an edge. This may also be useful for dynamical systems (e.g., the mujoco ant navigating a maze). Are there limitations for the method on such problems, e.g., edges may no longer be symmetric?
- How does SPTM compare when the space has been explored already?
- Can more quantitative results been shown such as solution path cost?
- Provide definitions for Fidelity, feasibility, and completeness and the source of data (polling human’s) in the Table 1 caption.
- “As shown in Table 5.2, “, should be renamed to Table 1.


**Experience Assessment:**

I have published one or two papers in this area.

**Review Assessment: Checking Correctness Of Derivations And Theory:**

I assessed the sensibility of the derivations and theory.

**Review Assessment: Checking Correctness Of Experiments:**

I carefully checked the experiments.

**Review Assessment: Thoroughness In Paper Reading:**

I read the paper thoroughly.

---

> ### Author Response · Authors · 2019-11-15
> **Response to Reviewer #3**
>
> Dear Reviewer #3,
>
> Thank you very much for your constructive feedback.
>
> As mentioned in your review, we recognize that our paper has only covered a small subset of possible experiments possible when testing visual planning (VP) problems.
>
> More difficult experiments: Actually, one contribution of our work is investigating what ‘difficult’ exactly means in the context of visual planning. As we show, methods such as SPTM and visual MPC, which were demonstrated on seemingly more complex tasks that involve first-person navigation or real images, fail on the simple tasks we investigate. This, at the very least, requires us to better think about the different difficulty axes in visual planning. Concretely, along with tackling different view-points and visual clutter, there is the difficulty of understanding the planning problem from an image, and solving it; our work addresses the latter.
>
> That said, following some of the reviewer’s suggestions, we have added an additional experiment to show the generality of our approach. In this experiment, an object can be  translated and rotated (SE(2)) slightly per timestep. Different from the experiments in the paper, here we change the object shape (and give the object shape as a context image). The goal is to plan a manipulation of an unseen object through a narrow gap between obstacles in zero-shot. Thus, our algorithm has to learn how object geometry is related to movement between obstacles, and has to generalize this knowledge for planning. Of course, all this has to be learned only from raw images. Please see our general comment for more details and additional results.
>
> Asymmetric transitions: Our framework does not assume symmetric transitions. We are using a directed graph, and the bilinear weight matrix in our energy cost function is not symmetric.
>
> SPTM on already explored space: If we were to test SPTM vs. HTM when the space has been explored already, then the only difference between the two models would be (1) the classifier and (2) the method of edge weighting during planning. In Figure 3 (Right), we show that SPTM (vanilla classifier + binary edge weighting) quantitatively averages to about 1.8 L2 distance from the goal, where as our method (CPC energy model + inverse of norm edge weighting) averages to about 0.4 L2 distance.
>
> Note in our experiment we actually test the SPTM and HTM on the same samples from the CVAE. We find that HTM chooses higher fidelity images and the plans are significantly more feasible. This results in a higher execution success rate.
>
> Solution path cost: Measuring the cost of visual plans is indeed an interesting question. One could measure the number of subgoals in the plans. However, an algorithm can output no subgoals at all and claim the shortest plan, or output many small steps to make sure that the low-level policy can follow. This is a tricky problem, and we defer it to human evaluation in this work. We find that the cost seems to be a function of feasibility, fidelity, and completeness given a low-level policy.
>
> Definitions for fidelity,etc: As requested, we will provide definitions of fidelity, feasibility, and completeness in Table 1 along with the source of the data.

---

### Author Response · Authors · 2019-11-15
**Additional Experimental Results**

Following the reviewers' feedback, we have added an additional experiment to show the generality of our approach. In this experiment, an object can be  translated and rotated (SE(2)) slightly per timestep. Different from the experiments in the paper, here we change the object shape (and give the object shape as a context image). The goal is to plan a manipulation of an unseen object through a narrow gap between obstacles in zero-shot. Thus, our algorithm has to learn how object geometry is related to movement between obstacles, and has to generalize this knowledge for planning. Of course, all this has to be learned only from raw images.

Due to time constraints of the rebuttal period, we were not able to perform a full evaluation, but we present encouraging preliminary results which are reflected in the resubmitted PDF in the Appendix. We trained a CPC energy model on the domain using the object shape as a context image, as shown in Figure 6 of the Appendix. We subsequently ran visual planning on samples collected in the test environment (similar to the SPTM setting) for shapes that were not seen during CPC training. We compare the CPC model with an SPTM-style classifier as a baseline. Note that the CPC energy model is able to generate smooth plans that mimic the proper rotational and movement constraints of unseen objects (see Figure 7). On the other hand, the plans produced by SPTM fail to assume such properties of the new object and often jump around (see Figure 8). We also present preliminary results for HTM, with a CVAE trained conditioned on the shape. The results of the CVAE hallucinated plans can be seen in Figure 9. Although the CVAE was unable to finish training, the results clearly show that HTM in a zero-shot generalization setting is able to generate a successful plan that rotates the object correctly in order to pass through a narrow opening.

---

### Decision · Program_Chairs · 2019-12-19

**Decision:**

Reject

**Comment:**

The submission presents an approach to visual planning. The work builds on semi-parametric topological memory (SPTM) and introduces ideas that facilitate zero-shot generalization to new environments. The reviews are split. While the ideas are generally perceived as interesting, there are significant concerns about presentation and experimental evaluation. In particular, the work is evaluated in extremely simple environments and scenarios that do not match the experimental settings of other comparable works in this area. The paper was discussed and all reviewers expressed their views following the authors' responses and revision. In particular, R1 posted a detailed justification of their recommendation to reject the paper. The AC agrees that the paper is not ready for publication in a first-tier venue. The AC recommends that the authors seriously consider R1's recommendations.